# The Association between Smoking Cessation and Depressive Symptoms: Diet Quality Plays a Mediating Role

**DOI:** 10.3390/nu14153047

**Published:** 2022-07-25

**Authors:** Shuo Liu, Hongbin Jiang, Dongfeng Zhang, Jia Luo, Hua Zhang

**Affiliations:** 1Department of Epidemiology and Health Statistics, Qingdao University Medical College, Qingdao 266000, China; liushuo0109@163.com (S.L.); zhangdf1961@126.com (D.Z.); luojia9605@163.com (J.L.); 2Department of Primary Care, Qingdao Municipal Hospital, Qingdao 266000, China; jhbsly@126.com; 3Department of Epidemiology and Health Statistics, The College of Public Health of Qingdao University, Qingdao 266000, China; 4Municipal Centre of Disease Control and Prevention of Qingdao, Qingdao 266034, China

**Keywords:** depressive symptoms, Healthy Eating Index, smoking cessation, NHANES, cross-sectional study

## Abstract

Objective: This cross-sectional study aimed to explore the association between smoking cessation and depressive symptoms and investigate the mediating role of dietary quality. Methods: We used data from the 2007–2014 National Health and Nutrition Examination Survey. Logistic regression models were applied to evaluate the associations between smoking cessation and depressive symptoms. Stratified analysis was performed according to different HEI levels. We examined the mediating role of HEI in the relationship between depressive symptoms and cessation duration using the Karlson–Holm–Breen (KHB) method. Results: A total of 20,004 participants aged 20 years or older were included in the analyses. There were significant correlations between years for smoking cessation and depressive symptoms (OR: 0.985, 95% CI: 0.971~0.999) after adjusting for correlation covariables. A likelihood ratio test showed that there was an interaction between smoking cessation and diet quality (*p* = 0.047). In the mediation analysis, we estimated that the increase in HEI scores after quitting smoking could explain the 6.91% decline in depressive symptoms. Conclusion: In this cross-sectional study, smoking cessation showed a protective effect on depressive symptoms and that diet quality can influence and mediate this association.

## 1. Introduction

Smoking is a major global public health problem. Nearly 8 million people died in 2019 as a result of smoking, and the number of smokers around the world continues to increase [1]. Smoking is a major risk factor for the development of cancer and cardiovascular and respiratory diseases [2], and quitting smoking can significantly reduce these health risks [3]. However, the link between smoking cessation and mental illness, such as depressive symptoms, is controversial. Many qualitative and quantitative studies reported quitting smoking can effectively improve mental health [4,5,6,7,8,9,10]. However, many smokers believe that smoking can help them relieve stress and reduce anxiety or that quitting smoking can exacerbate mental illness. These beliefs lead smokers to give up on quitting [11,12]. Exploring the link between smoking and depressive symptoms is meaningful because it could lead to examining whether people try to quit and whether they are successful.

At present, the focus of epidemiological research has shifted to understanding the potential complexity of the relationships among behavioral risk factors. This includes the relationship between smoking status and dietary exposure [13,14,15]. Some studies evaluating nutritional habits have shown that quitters have a better diet than smokers. In general, quitters have a higher intake of fruits, vegetables, grains, etc. and a lower intake of alcohol, total energy, and total fat [16,17,18,19]. Cross-sectional studies and randomized, controlled trial studies have found an association between smoking cessation behavior and higher diet quality [20,21,22]. Previous studies have also linked dietary quality to depressive symptoms [23,24,25,26,27]. A meta-analysis of observational studies showed a negative association between HEI and the prevalence of depression (longitudinal: OR: 0.76, 95% CI: 0.57, 1.02; cross-sectional study: OR = 0.53, 95% CI: 0.38, 0.75) [25]. Clinical intervention studies also provide relevant evidence [26,27]. A meta-analysis of 16 RCTS reported the outcome data from 45,826 individuals, showing a significant reduction in depressive symptoms with a small combined effect compared with the control conditions (g = 0.275, 95% CI = 0.10 to 0.45, *p* = 0.002) [27]. In conclusion, diet quality is associated with both smoking cessation and depressive symptoms. However, few studies have explored whether dietary quality mediates the relationship between quitting smoking and depressive symptoms.

From a public health perspective, a better understanding of the relationship between smoking cessation status and depressive symptoms, as well as the role of dietary quality in this relationship, can improve the success rate of smoking cessation in patients with depressive symptoms and guide reasonable and effective smoking cessation strategies.

## 2. Method

### 2.1. Study Population

The NHANES is a cross-sectional survey of the community-dwelling US population. Since 1999, the data collection cycle used has been 2 years. Personal interviews and standardized medical examinations are included in each data collection [28]. The NHANES’s study was approved by the Ethics Review Board of the National Center for Health Statistics. Written informed consent was obtained from the participants [29].

We extracted data from NHANES 2007–2018, 2009–2010, 2011–2012, and 2013–2014 (four cycles), and a total of 40,617 people were initially included in our study. The cycle was chosen based on the completeness of the data we needed for each cycle and the consistency of the collection method. Our analyses were limited to participants 18 years and older, with 24,732 people in all. From the remaining participants, we excluded participants with incomplete depression questionnaire data (*n* = 3474) or smoking status data (*n* = 883). Females who were pregnant or breastfeeding (*n* = 307) were also excluded. We further excluded 64 individuals who reported extreme total energy intakes (500 or 5000 kcal/day for women and 500 or 8000 kcal/day for men). Finally, 20,004 participants were included in our study (10,007 males and 9997 females) (Figure 1).

### 2.2. Depressive Symptoms Assessment

In the 2007–2014 NHANES, depressive symptoms were assessed using the Patient Health Questionnaire-9 (PHQ-9). It asked respondents how many days in the past two weeks they had experienced symptoms of depression [30]. PHQ-9 scores range from 0 to 27. We used a total score of ≥10 as a cut point for the presence of depressive symptoms. The PHQ-9 has 88% sensitivity and 88% specificity for the diagnosis of major depression for scores of 10 or more [31].

### 2.3. Classification of Smoking Status

We assessed smokers’ statuses using self-reported questionnaires. A never smoker is defined as someone who has smoked less than 100 cigarettes in their lifetime. Those who had smoked more than 100 cigarettes in their lifetime were further divided by asking, “Are you a current smoker?” If the respondent replied that he or she did not currently smoke, he or she was classified as a “quitter”. If the respondent answered “yes” to this question, then he or she was classified as a “current smoker”. For the quitters, we uniformly converted the time of cessation to the variable “year of cessation” as the exposure for regression analysis. Furthermore, the time to quit smoking as a categorical variable was classified as less than 10 years, 10–20 years, 20–30 years, and over 30 years.

### 2.4. Dietary Quality Assessment

Our study aimed to explore the mediating role of diet in the relationship between smoking cessation and depressive symptoms. Measuring the overall quality of a diet, as a popular alternative, can provide more information to reflect the multifaceted nature of an individual’s diet and therefore better study our research objectives. We assessed dietary quality using the Healthy Eating Index (HEI)-2015. The HEI-2015 is based on the Dietary Guidelines for Americans (DGA) for 2015–2020 [32]. It is made up of 13 components that are summed up to produce a 100-point total. The total score is the sum of the score of the adequacy components (i.e., foods to eat more of for good health) and moderation components (i.e., foods to limit for good health) [33]. For further stratified analysis, HEI-2015 scores less than 50, between 50 and 70, and more than 70 were classified as inadequate, average, and optimal, respectively [34,35].

### 2.5. Covariates

The following covariates were examined as potential confounders of the association between smoking cessation and depressive symptoms: sex, age, race, the degree of education, income, alcohol use, total energy and caffeine intake, diabetes, hypertension, HEI scores, work activity intensity, recreational activity intensity, and BMI. These covariates were chosen based on previous experience and similar studies [36,37] Details about these covariates are provided in Appendix A.

### 2.6. Statistical Analyses

We used R Studio (R Studio, Boston, MA, USA) and STATA 15.0 (Stata Corporation, College Station, TX, USA) to perform all analyses. The NHANES dataset includes the sampling weights, stratifications, and clusters that were incorporated into the analysis in order to obtain the proper estimates and standard errors. New sample weights were created for the combined NHANES cycles (2007–2014) according to the analysis guidelines for the continuous NHANES.

We first conducted descriptive analyses. For continuously normally distributed variables, we used the mean ± standard deviation, and for the non-normally distributed variables, the median and interquartile range were computed. Categorical variables were expressed as numbers and percentages. The general characteristics (normal and non-normal variables) of participants with or without depressive symptoms were compared using the student’s *t*-test or nonparametric tests. A comparison between the groups for categorical variables was performed using the chi-square test.

Associations between the smoking cessation status and depressive symptoms were evaluated using univariate and multivariate logistic regression analysis. The time to quit smoking was placed in the model as continuous variables and multi-classification variables, respectively. Model 1 was adjusted for age and sex. All covariates were included in model 2. Odds ratios (OR) and a 95% CI were calculated to test the relative risk for association. The possible nonlinear relationship between the duration of smoking cessation and depressive symptoms was evaluated with the restricted cubic spline regression. We stratified the analysis with the HEI levels to determine whether the relationship between smoking cessation and depressive symptoms depended on diet quality. We fitted the restricted cubic splines according to different levels. Testing the significance of the interaction effects, we used likelihood ratio tests comparing the models with and without the interaction terms of time since quitting smoking variables and HEI scores.

Next, Baron and Kenny’s causal step method explored the relationship between cessation duration, diet quality, and depressive symptoms [38]. In this method, X represents the independent variable (year of cessation), Y represents the dependent variable (depressive symptoms), and M stands for the medium (diet quality). Finally, we examined the mediating role of the HEI in the relationship between depressive symptoms and cessation duration using the Karlson–Holm–Breen (KHB) method. The method breaks down the total effect of the variables into direct and indirect effects, and finally the percentage of mediation is determined.

## 3. Results

### 3.1. Descriptive Analysis of the Study Participants

Table 1 shows the sociodemographic characteristics of the analytic sample. Of the 20,004 participants in this study, the prevalence of depressive symptoms (PHQ-9 scale ≥ 10) was 9.52%. Participants with depressive symptoms were more likely to be middle-aged, female, drinkers, obese people, have lower HEIs, have less education, have less family income, and have lower work activity intensities and recreational activity intensities, have lower total energy intakes, have higher caffeine intakes and suffer from hypertension, and have diabetes.

### 3.2. Smoking Cessation, HEI, and Depressive Symptoms

We subsequently evaluated the association between the time since quitting smoking and depressive symptoms. Table 2 presents the results of the analytic models. In the crude model, the duration of cessation was a protective factor for depressive symptoms (OR: 0.983, 95% CI: 0.973~0.995). The results were consistent in model 1 (OR: 0.977, 95% CI: 0.954~0.990) and model 2 (OR: 0.985, 95% CI: 0.971~0.999). When several levels of the duration of smoking cessation were analyzed as categorical variables, the protective effect of each level was also significant.

Differences in the HEI altered the association between smoking cessation and depressive symptoms (interaction *p* value = 0.047). The results of the stratified analysis confirmed the existence of the interaction. The detailed results are shown in Appendix A. The duration of smoking cessation was significantly associated with depressive symptoms in the inadequate HEI (OR: 0.974, 95% CI: 0.952~0.996) and average groups (OR: 0.980, 95%CI: 0.960~0.999). However, in the optimal group, the correlation was not significant (OR: 1.031, 95% CI: 0.997~1.037). The results did not change when exposure was analyzed as a categorical variable.

The dose–response relationship between the duration of smoking cessation and depressive symptoms is shown in Figure 2A. Depressive symptoms showed a monotonous downward trend as the time to quit smoking increased. In stratified analyses, similar dose–response trends were observed in both the inadequate HEI and average groups (Figure 2B,C). However, in the optimal group, cubic spline analyses showed inconsistent results, and the results were not significant (Figure 2D).

We used model 2 for analysis. The dashed lines represent the 95% confidence intervals. Figure 2A represents the whole research population. Figure 2B–D represent the inadequate HEI group, the average group, and the optimal group, respectively.

The results of the causal steps method are presented in Appendix A. For path a, the duration of cessation was positively correlated with the HEI scores (β = 0.118, *p* < 0.05). For path b, after adjusting for all the covariates as well as the years since smoking cessation, the HEI scores were related to a lower risk of depressive symptoms (β = −0.016, *p* < 0.05). For path c, the years since smoking cessation showed a protective effect on depressive symptoms (β = −0.018, *p* < 0.05). After introducing the HEI scores into the models (path c’), the regression coefficient between the years since smoking cessation and depressive symptoms was slightly reduced (β = −0.017, *p* < 0.05), suggesting a partially mediating effect of dietary quality on the relationship between smoking cessation duration and depressive symptoms. In Table 3, we present the results of the mediational analysis. After adjusting for all the covariates, the total effect of quitting smoking on depressive symptoms was −0.018 (95% CI = −0.028~−0.008), and the indirect effect of quitting smoking through the HEI was −0.001 (95% CI = −0.028~−0.008), with 6.91% of the total effect being mediated.

## 4. Discussion

In this study, a total of 20,004 participants meeting the inclusion criteria were identified in the NHANES database (2007–2014). We observed that quitting smoking was significantly associated with a reduction in depressive symptoms, and the risk of depressive symptoms continued to decrease as the duration of quitting increased. We also found an interaction between the cessation duration and Healthy Eating Index (HEI) (interaction *p* value = 0.047). In stratified analysis, the cessation duration was significantly associated with depressive symptoms in the inadequate HEI and average groups but not in the optimal HEI group. The causal step method showed that the years of cessation were positively associated with the HEI and that the HEI was associated with a reduced risk of depressive symptoms. When the HEI was added to the regression equation for years of quitting and depressive symptoms, the beta was slightly reduced. The results suggest that the HEI played a partially mediating role in the relationship between the duration of smoking cessation and depressive symptoms. The KHB method showed that 6.91% of the total effect was mediated.

Previous studies examining the link between quitting smoking and depressive symptoms have generally found that quitting has a positive effect. In a randomized clinical trial in which the participants were heavy drinkers and smokers, the results showed a protective effect on the development of depressive symptoms [8]. Another study found similar results. Smokers were given an intervention to quit smoking and were followed up on for 2 years. The results found a 32.9 percent reduction in depressive symptoms among those who had quit smoking compared with those who continued to smoke [39]. However, in other studies, there have been conflicting conclusions. Smoking cessation can result in the new onset of major depressive disorders, regardless of whether the individual has a history of depression or not [40]. Quitting smoking was not associated with depressive symptoms [41]. Alternatively, how depression changes after quitting smoking may depend on the effect of the medication [42].

Our study suggests that dietary quality may influence and mediate the association between quitting smoking and the risk of depressive symptoms. According to the results of the causal step method, the years of quitting smoking were positively correlated with the HEI. Our results are consistent with those of previous studies. A cross-sectional study based on the Cardiovascular Risk Factors in Luxembourg (ORISCAV-LUX) survey explored the relationship between smoking and overall diet quality. The study examined the associations between eight diet quality indexes and found that former smokers had higher overall diet quality indexes than current smokers of varying intensities [20]. A randomized controlled trial of patients with head and neck cancer produced consistent results. After 12 months, the patients in the intervention group offered smoking cessation treatment had a significant improvement in the quality of their diets [21]. Similarly, the results from a multi-risk intervention trial showed that men who quit smoking showed greater beneficial changes in their dietary patterns, and early cessation showed the greatest improvement in all subgroups [22]. Meanwhile, previous studies have linked high dietary quality to a reduction in depressive symptoms. Leda Chatzi et al. used data from the Rhea study, a prospective mother-child cohort, to show that a healthy diet during pregnancy is associated with a reduced risk of depressive symptoms [43]. In addition, several cross-sectional studies examining the relationship between diet and depression have reported similar conclusions [44,45,46]. Randomized controlled studies have confirmed a causal relationship between dietary quality and depressive symptoms [26,27]. Although dietary quality is strongly associated with both quitting smoking and depressive symptoms, the studies that link the three of them have been fairly rare. To our knowledge, this is the first study to investigate the mediating role of dietary quality in the relationship between quitting smoking and depressive symptoms.

Our study has some advantages. First, we used a sample based on the national population, providing sufficient data and representation for our analysis. Moreover, we used the Health Substitution Index (HEI) to measure the overall quality of diets rather than focusing on individual foods. This could provide more information to reflect the multifaceted nature of human diets and thus better assess the relationship between diet and research factors. Despite these advantages, there are some limitations that should be noted. Frist, our study is limited by its cross-sectional design, which limits causal inferences. In addition, the participants’ smoking cessation data and diet data were self-reported and could be susceptible to reporting bias or recall bias. Furthermore, there were missing values of about 15% in the depression questionnaire data. We assessed the differences between those who provided depression data and those who did not and found that those not providing these data were older, less educated, and more likely to be male (Appendix A). This suggests that there may be systematic bias in our findings. Finally, although we adjusted for many covariates, the possibility of residual confounding cannot be entirely ruled out.

In summary, this study found that quitting smoking was significantly associated with a reduced risk of depressive symptoms, and the association was partly mediated by diet quality. The implications of this study are to confirm the beneficial effect of quitting smoking on depressive symptoms. This is used to stimulate and promote the smoking cessation behavior of depressed people. Moreover, prevention actions should be targeted primarily at people with low dietary quality. Tobacco control should also aim at promoting better nutritional habits. It seems worthwhile to consider a holistic approach to prevention.

## Figures and Tables

**Figure 1 nutrients-14-03047-f001:**
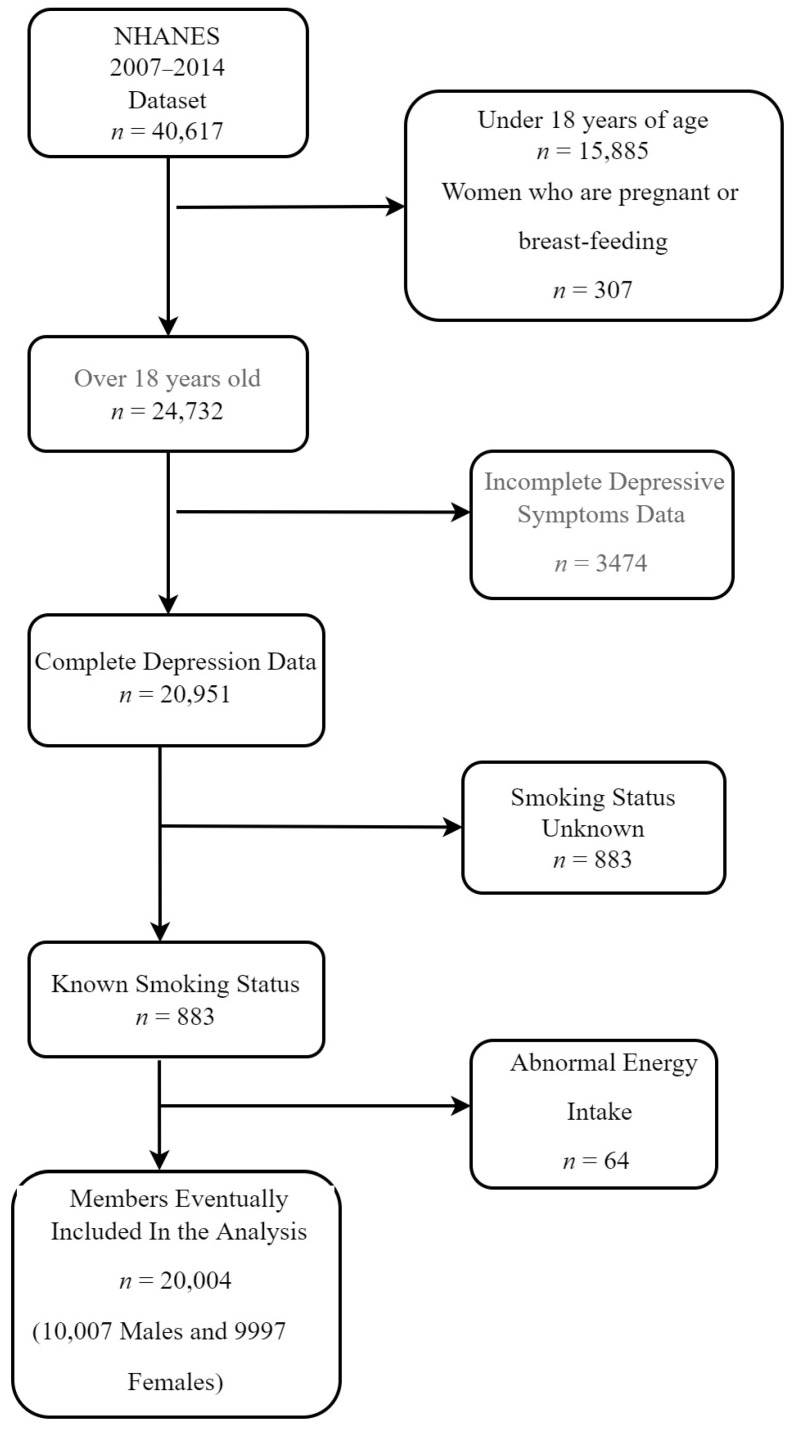
Flow diagram of the selection of eligible participants for NHANES 2007–2014.

**Figure 2 nutrients-14-03047-f002:**
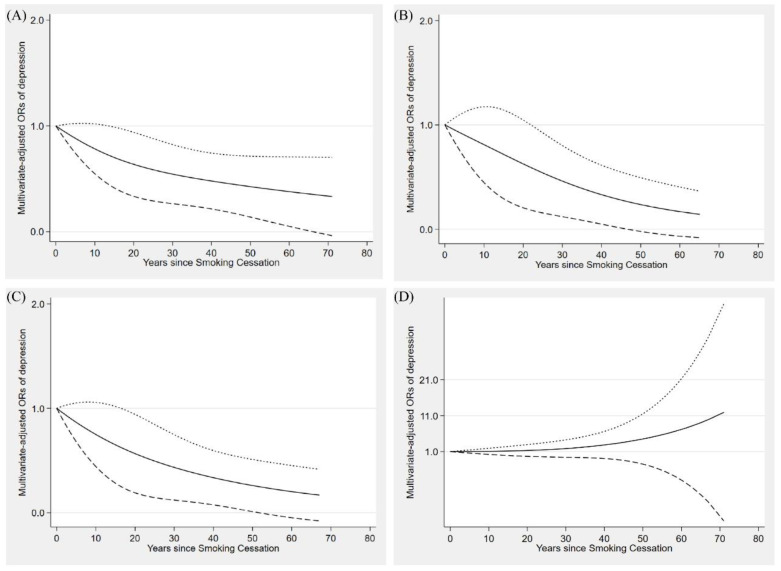
Restricted cubic spline model of the odds ratios of depression symptoms with years since smoking cessation. Adjusted for age, gender, body mass index, race, education, family income, alcohol consumption, work activity, recreational activity, hypertension, diabetes, and daily total energy and coffee intake. The dashed lines represent the 95% confidence intervals. (**A**) represents the whole research population. (**B**–**D**) respectively represent the inadequate HEI group, the average group and the optimal group.

**Table 1 nutrients-14-03047-t001:** Baseline characteristics of participants by depressive symptoms under NHANES 2007–2014 (*n* = 20,004).

	Non-Depressive Symptoms (PHQ < 10)	Depressive Symptoms (PHQ ≥ 10)	*p* Value
Number of participants (%) ^a^	18,100 (90.48)	1904 (9.52)	
Age	50 (35, 64)	50 (37, 62)	0.001
Age (year) ^a^			<0.001
18–39	6124 (36.07)	590 (34.15)	
40–59	5818 (37.76)	772 (45.10)	
≥60	6158 (21.17)	542 (20.75)	
Gender (%) ^a^			<0.001
Male	9334 (50.91)	673 (34.85)	
Female	8766 (49.09)	1231 (65.15)	
Race or ethnicity (%) ^a^			<0.001
Mexican American	2677 (8.28)	290 (8.82)	
Other Hispanic	1761 (5.13)	266 (8.12)	
Non-Hispanic White	8125 (69.22)	824 (63.70)	
Non-Hispanic Black	3798 (10.77)	408 (13.98)	
Other races	1739 (6.61)	116 (5.97)	
Educational level (%) ^a^			<0.001
<High school	1704 (4.89)	286 (9.10)	
High school	6638 (33.18)	884 (46.14)	
>High school	9446 (61.93)	713 (44.76)	
Household income (%) ^a^			<0.001
Under USD 20,000	3509 (13.35)	721 (30.28)	
USD 20,000 and over	13,854 (86.65)	1090 (69.72)	
Body mass index ^b^	27.80(24.20, 32.20)	29.77(25.09, 34.98)	
Body mass index (%) ^a^			<0.001
<18.5 kg/m^2^	275 (1.42)	38 (1.92)	
18.5–25 kg/m^2^	5107 (29.16)	425 (23.50)	
25–30 kg/m^2^	6057 (33.82)	497 (27.11)	
≥30 kg/m^2^	6661 (35.59)	944 (47.48)	
Work activity (%) ^a^			0.0012
Vigorous	3372 (20.49)	327 (18.54)	
Moderate	3880 (23.35)	343 (19.05)	
Other	10,846 (56.17)	1232 (62.41)	
Recreational activity (%) ^a^			<0.001
Vigorous	4047 (26.28)	180 (9.18)	
Moderate	4942 (29.32)	359 (21.87)	
Other	9110 (44.39)	1365 (68.95)	
Alcohol consumption (%) ^a^	13,054 (77.71)	1361 (75.73)	0.1039
Smoke status (%) ^a^			<0.001
Never smoker	10,181 (56.46)	791 (39.51)	
Former smoker	4378 (24.55)	408 (21.01)	
Current smoker	3540 (19.00)	715 (39.48)	
Healthy Eating Index ^b^	55.52 (44.37, 63.20)	49.56 (41.18, 58.88)	
Healthy Eating Index (%) ^a^			
<50	7031 (40.72)	940 (53.62)	
50–70	8245 (46.39)	755 (40.08)	
≥70	2241 (12.88)	131 (6.30)	
Diabetes (%) ^a^	3214 (13.54)	511 (21.45)	<0.001
Hypertension (%) ^a^	9766 (49.83)	1162 (56.94)	<0.001
Total energy (kcal/day) ^b^	1907.5 (1466, 2462)	1799.5 (1360.5, 2354)	<0.001
Caffeine intake (mg/day) ^b^	96.5 (28.5, 200)	101 (32.221)	0.0051

Data are number of participants (weighted percentage) or medians (interquartile ranges). PHQ = Patient Health Questionnaire. ^a^ Chi-square test was used to compare the percentage between participants with and without depression symptoms. ^b^ Mann–Whitney U test was used to compare the difference between participants with and without depression symptoms. USD: U.S. dollar.

**Table 2 nutrients-14-03047-t002:** Association between smoking cessation and depressive symptoms, adjusting for several confounders, NHANES 2007–2014 (*n* = 20,004).

Variable	Crude Model		Model 1		Model 2	
	OR	95% CI	OR	95% CI	OR	95% CI
Time since quitting (years)	0.983	0.973–0.995	0.977	0.954–0.990	0.985	0.971–0.999
Smoking status						
Current smokers	1	(ref)	1	(ref)	1	(ref)
Time since quitting (years)						
<10 years	0.524	0.409–0.671	0.524	0.411–0.668	0.636	0.485–0.833
10–20 years	0.360	0.274–0.474	0.349	0.263–0.464	0.364	0.264–0.503
20–30 years	0.389	0.258–0.586	0.385	0.251–0.591	0.563	0.357–0.889
≥30 years	0.308	0.217–0.438	0.305	0.202–0.459	0.449	0.290–0.695
Never smoked	0.337	0.295–0.384	0.306	0.265–0.354	0.433	0.363–0.518
Age (years)			1.00	0.996–1.006	0.987	0.981–0.994
Sex						
Male			1	(ref)	1	(ref)
Female			2.127	1.855–2.439	1.876	1.542
Race						
Mexican American					1	(ref)
Other Hispanic					1.630	1.219–2.107
Non-Hispanic White					1.100	0.913–1.327
Non-Hispanic Black					1.133	0.915–1.405
Other Race					1.443	1.086–1.919
Education						
Below high school					1	(ref)
High school level					0.834	0.686–1.017
Above high school					0.678	0.540–0.853
Annual household income						
Less than USD 20,000					1	(ref)
Over USD 20,000					0.489	0.424–0.565
Caffeine intake (mg)					1.000	1.000–1.000
Energy intake (kcal)					0.999	0.999–1.000
Drinking status						
Drinkers					1.195	1.009–1.415
Non-drinkers					1	(ref)
Hypertension status						
Hypertension					1.277	1.050–1.554
No hypertension					1	(ref)
Diabetes status						
Diabetes					1.477	1.202–1.813
No diabetes					1	(ref)
Intensity of work activities						
Vigorous					1	(ref)
Moderate					1.739	1.328–2.278
Mile					2.681	2.121–3.412
Intensity of recreational activities						
Vigorous					1	(ref)
Moderate					0.861	0.679–1.092
Mild					1.149	0.938–1.406
Body Mass Index (kg/m^2^)					1.022	1.013–1.032
Healthy Eating Index					0.987	0.982–0.993

Calculated using binary logistic regression.

**Table 3 nutrients-14-03047-t003:** Direct and indirect effect of smoking cessation and mediators on depression symptoms.

Effect	β	SE	Z	*p*	95% CI		Mediation (%)
					Lower	Upper	
Depression symptoms							
Total effect	−0.017	0.005	−3.32	<0.01	−0.027	−0.007	
Direct effect	−0.018	0.005	−3.46	<0.01	−0.028	−0.008	
Indirect effect	−0.001	0.001	−2.08	0.037	−0.024	−0.001	6.91

SE = standard error.

## Data Availability

The data that support the findings of this study are openly available from the National Health and Nutrition Examination Survey (https://www.cdc.gov/nchs/nhanes/index.htm), (accessed on 3 May 2022).

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
