# Peer review of "The Association between Smoking Cessation and Depressive Symptoms: Diet Quality Plays a Mediating Role"

_nutrients, 2022, doi:10.3390/nu14153047_

Round 1

Reviewer 1 Report

Introduction

 These numbers appear to be a mistake (lines 33-36):

1Smoking is a major risk factor for the development of cancer, cardiovascular and respiratory diseases,2 and quitting smoking can significantly reduce these health risks.3 But the link between smoking cessation and mental illness, such as depressive symptoms, is controversial

The literature cited in the introduction regarding the diet-mental health link is insufficient (and certainly the cross-sectional NHANES study is not the sentinel paper in this very wide field). It should cite at least one of the meta-analyses concerning the link between diet quality and the risk for depression (e.g. https://www.ncbi.nlm.nih.gov/pubmed/30254236 ) and at least one of the randomised controlled trials (e.g. https://www.ncbi.nlm.nih.gov/pubmed/28137247 ) and meta-analysis on dietary improvement leading to depressive symptoms (https://www.ncbi.nlm.nih.gov/pubmed/30720698).

(Be aware that a recent paper Shafiei F, Salari-Moghaddam A, Larijani B, Esmaillzadeh A. Adherence to the Mediterranean diet and risk of depression: a systematic review and updated meta-analysis of observational studies. Nutrition reviews. 2019 Apr 1;77(4):230-9. presents incorrect data and conclusions and should not be cited).

Methods:

Data from the NHANES 2007-2014 were used, but it’s not clear how many data collection cycles this includes, or why these ones were selected. Importantly, the reason for so many missing data on depression needs to be clarified, and an analysis comparing the demographics of those who did and did not answer the depression questions conducted; there might be systematic bias in the missing data.

It was not made clear in the original aims that diet quality would be investigated as a potential effect modifier as well as a mediator – these are two different concepts (see further comments below).

I’m concerned by the inclusion of so many of the potential confounders at step 2, without consideration of the possibility of understanding further what variables were attenuating the main effect. A DAG would have helped to guide the model design and likely avoided the need to include such a large number of variables.

Age – more likely to be middle-aged rather than younger (or older)?

Why is Figure 2 in the main article, but Figure 1 in the supplementary? I think Fig 1 should be in the main results

I’m very confused by the results cited in lines 192-195– according to what was stated, the inclusion of HEI did not have an impact on the main effect of smoking cessation and depressive symptoms (beta was almost unchanged – in fact, slightly strengthened).

Discussion

Lines 209-210 – track changes left in text

I’m concerned by this:

‘In stratified analysis, cessation duration was significantly associated with depressive symptoms in the HEI 2 inadequat and average groups, but not in the optimal HEI group. These results suggest that dietary quality may mediate the relationship between cessation duration and depressive symptoms.’

This is incorrect – what is being discussed is HEI as an effect modifier – i.e. the association between smoking cessation and depressive symptoms is different for different levels of HEI.

‘Mediation’ is when the possible impact of the exposure variable (smoking cessation) works through a variable on the causal pathway.

This means that the mediation hypothesis is that smoking cessation LEADS TO a healthier diet, which LEADS TO reduced depression. There is little no evidence for this, yet the title and discussion suggest that there is.

Apart from anything else, there is also no evidence to allow us to understand whether smoking cessation leads to increases in diet quality or vice versa – the fact that health behaviours tend to be associated does not mean that one leads to the other, or that we can clarify the temporality of any causal relationship without a focused epidemiological study to test this. It’s likely to vary between individuals. However, the literature presented in the discussion should be at least touched on in the introduction as a rationale for the study.

Again, the literature on the field of research linking diet quality to reduced depression risk presented in the discussion is very limited and not informed by the latest evidence.

Reviewer 2 Report

the work is very interesting and well written. however I suggest some changes:

insert more details in point 2.4 useful for arguing in more detail what the goal of the work is.

expand, both in the results and in discussion, the information relating to food and consequently to the quality of the diet in relation to depressive symptoms such as the title states

Round 2

Reviewer 1 Report

Comments 4: Importantly, the reason for so many missing data on depression needs to be clarified, and an analysis comparing the demographics of those who did and did not answer the depression questions conducted; there might be systematic bias in the missing data.

Reply: Thank you very much for your comments. Approximately 15% of our participants were excluded due to the absence of depression Score scale data.  The Depression Scale was collected using MEC interview methods. The sample of people examined by MEC is a subset of the respondents in the survey. In addition, we suspect that sensitive data may face more lost data. We compared the basic characteristics of members with complete data and missing data, and the results are as follows. We add the systematic bias generated by this part to the “limitation”.

This information is not properly reflected in the discussion. You state “Furthermore, we excluded nearly 15% of the missing depressive symptom data, which may also lead to systemic bias.”

Better would be: ‘We assessed differences between those who provided depression data and those who did not and found that those not providing these data were older, less educated, and more likely to be male (data not shown). This suggests that there may be systematic bias in our findings.’

Results

Fig S1 in the supplementary is Fig2 in the manuscript?

Where is Fig S1 with the results of the causal steps method?

In the discussion:

 One cohort study identified that greater adherence to a healthy diet was associated with a reduced likelihood of depressive symptoms [43] and three cross-sectional studies have similarly reported this association [44-46].”

There are a great many more than one cohort and three cross-sectional studies – this statement is very misleading. Please expand to reflect the literature as it is.
